# GENERATIVE ADVERSARIAL PARALLELIZATION

**Daniel Jiwoong Im**
AIFounded Inc.
Toronto, ON
{daniel.im}@aifounded.com

**He Ma, Chris Dongjoo Kim and Graham W. Taylor**
University of Guelph
Guelph, ON
{hma02,ckim07,gwtaylor}@uoguelph.ca

## ABSTRACT

Generative Adversarial Networks (GAN) have become one of the most studied frameworks for unsupervised learning due to their intuitive formulation. They have also been shown to be capable of generating convincing examples in limited domains, such as low-resolution images. However, they still prove difficult to train in practice and tend to ignore modes of the data generating distribution. Quantitatively capturing effects such as mode coverage and more generally the quality of the generative model still remain elusive. We propose Generative Adversarial Parallelization (GAP), a framework in which many GANs or their variants are trained simultaneously, exchanging their discriminators. This eliminates the tight coupling between a generator and discriminator, leading to improved convergence and improved coverage of modes. We also propose an improved variant of the recently proposed Generative Adversarial Metric and show how it can score individual GANs or their collections under the GAP model.

## 1 INTRODUCTION

The growing popularity Generative Adversarial Networks (GAN) and their variants stems from their success in producing realistic samples (Denton et al., 2015; Radford et al., 2015; Im et al., 2016; Salimans et al., 2016; Dumoulin et al., 2016) as well as the intuitive nature of the adversarial training framework (Goodfellow et al., 2014). Compared to other unsupervised learning paradigms, GANs have several merits:

- The objective function is not restricted to distances in input (e.g. pixel) space, for example, reconstruction error. Moreover, there is no restriction to certain type of functional forms such as having a Bernoulli or Gaussian output distribution.
- Compared to undirected probabilistic graphical models (Hinton et al., 2006; Salakhutdinov & Hinton, 2009), samples are generated in a single pass rather than iteratively. Moreover, the time to generate a sample is much less than recurrent models like PixelRNN (Oord et al., 2016).
- Unlike inverse transformation sampling models, the latent variable size is not restricted (Hyvarinen & Pajunen, 1999; Dinh et al., 2014).

In contrast, GANs are known to be difficult to train, especially as the data generating distribution becomes more complex. There have been some attempts to address this issue. For example, Salimans et al. (2016) propose several tricks such as feature matching and minibatch discrimination. In this work, we attempt to address training difficulty in a different way: extending two player generative adversarial games into a multi-player game. This amounts to training many GAN-like variants in parallel, periodically swapping their discriminators such that generator-discriminator coupling is reduced. Figure 1 provides a graphical depiction of our method.

Besides the training dilemma, from the point of view of density estimation, GANs possess very different characteristics compared to other probabilistic generative models. Most probabilistic models distribute the probability mass over the entire domain, whereas GAN by nature puts point-wise probability mass near the data. The question of whether this is desirable property or not is still an open question[1]. However, the primary concern of this property is that GAN may fail to allocate mass to

---

[1]Noting that the general view in ML is that there is nothing wrong with sampling from a degenerate distribution (Neal, 1998).

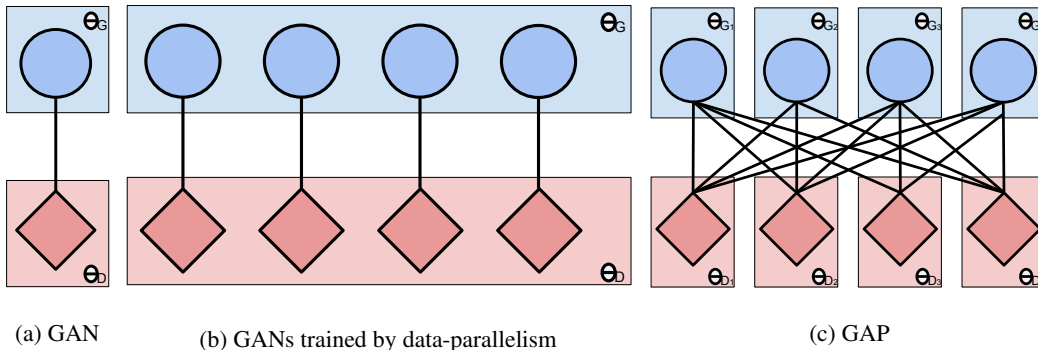

(a) GAN (b) GANs trained by data-parallelism (c) GAP

Figure 1: Depiction of GAN, Parallel GAN, and GAP. Not intended to be interpreted as a graphical model. The difference between Figure (b) and (c) is that typical data-based parallelization is based on multiple models which share parameters. In contrast, GAP requires multiple models with their own parameters which are structured in a bipartite formation.

some important modes of the data generating distribution. We argue that our proposed model could alleviate this problem.

That our solution involves training many pairs of generators and discriminators together is a product of the fact that deep learning algorithms and distributed systems have been co-evolving for some time. Hardware accelerators, specifically Graphics Processing Units, (GPUs) have played a fundamental role in advancing deep learning, in particular because deep architectures are so well suited to parallelism (Coates et al., 2013). Data-based parallelism distributes large datasets over disparate nodes. Model-based parallelism allows complex models to be split over nodes. In both cases, learning must account for the coordination and communication among processors. Our work leverages recent advances along these lines (Ma et al., 2016).

## 2 BACKGROUND

The concept of a *two player zero-sum game* is borrowed from *game theory* in order to train a generative adversarial network (Goodfellow et al., 2014). A GAN consists of a generator $G$ and discriminator $D$, both parameterized as feed-forward neural networks. The goal of the generator is to generate samples that fool the discriminator from thinking that those samples are from the data distribution $p(\boldsymbol{x})$, *ad interim* the discriminative network's goal is to not get tricked by the generator.

This view is formalized into a *minimax objective* such that the discriminator maximizes the expectation of its predictions while the generator minimizes the expectation of the discriminator's predictions,

$$\min_{\boldsymbol{\theta}_G} \max_{\boldsymbol{\theta}_D} V(D, G) = \min_{\boldsymbol{\theta}_G} \max_{\boldsymbol{\theta}_D} \Big[ \mathbb{E}_{\boldsymbol{x} \sim p_{\mathcal{D}}} \big[ \log D(\boldsymbol{x}) \big] + \mathbb{E}_{\boldsymbol{z} \sim p_{\mathcal{G}}} \big[ \log \big( 1 - D(G(\boldsymbol{z})) \big) \big] \Big]. \quad (1)$$

where $\boldsymbol{\theta}_G$ and $\boldsymbol{\theta}_D$ are the parameters (weights) of the neural networks, $p_{\mathcal{D}}$ is the data distribution, and $p_{\mathcal{G}}$ is the prior distribution of the generative network.

Proposition 2 in (Goodfellow et al., 2014) illustrates the ideal concept of the solution. For two player game, each network's gain of the utility (loss of the cost) ought to balance out the gain (loss) of the other network. In this scenario, the generator's distribution becomes the data distribution. Remark that when the objective function is convex, gradient-based training is guaranteed to converge to a saddle point.

### 2.1 EMPIRICAL OBSERVATIONS

The reality of training GANs is quite different from the ideal case due to the following reasons:

1. The discriminative and generative networks are bounded by a finite number of parameters, which limits their modeling capacity.

2. Practically speaking, the second term of the objective function in Equation 1 is a bottleneck early on in training, where the discriminator can perfectly distinguish the noisy samples coming from the generator. The argument of the log saturates and gradient will not flow to the generator.

3. The GAN objective function is known to be non-convex and it is defined over a high-dimensional space. This often results in failure of gradient-based training to converge.

The first issue comes from the nature of the modelling problem. Nevertheless, due to the expressiveness of deep neural networks, they have been shown empirically to be capable of generating natural images (Radford et al., 2015; Im et al., 2016) by adopting parameter-efficient convolutional architectures. The second issue is typically addressed by inverting the generator's minimization into the maximization formulation in Equation 1 accordingly,

$$\min_{\boldsymbol{\theta}_G} \log(1 - D(G(\boldsymbol{z}))) \to \max_{\boldsymbol{\theta}_G} \log(D(G(\boldsymbol{z}))). \tag{2}$$

This provides better gradient flow in the earlier stages of training (Goodfellow et al., 2014).

Although there have been cascades of success in image generation tasks using advanced GANs (Radford et al., 2015; Im et al., 2016; Salimans et al., 2016), all of them mention the problem of difficulty in training. For example, Radford et al. (2015) state that *the generator ... collapsing all samples to a single point ... is a common failure mode observed in GANs*. This scenario can occur when the generator allocates most of its probability mass to a single sample that the discriminator has difficulty learning. Empirically, convergence of the learning curve does not correspond to improved quality of samples coming from the GAN and vice-versa. This is primarily caused by the third issue mentioned above. Gradient-based optimization methods are only guaranteed to converge to a Nash Equilibrium for convex functions, whereas the loss surface of the neural networks used in GANs are highly non-convex and there is no guarantee that a Nash Equilibrium even exists.

## 3 PARALLELIZING GENERATIVE ADVERSARIAL NETWORKS

The subject of generative modeling with GANs has undergone intensive study, and model evaluation between various types of GANs is topic of increased interest and debate (Theis et al., 2015). Our work is inspired by the Generative Adversarial Metric (Im et al., 2016). The GAM enables us to quantitatively evaluate any pair of GANs. The core concept of the GAM is to swap one discriminator (generator) with the other discriminator (generator) during the test phase (see the pictorial example in Figure 8). The GAM concept can easily be extended from evaluation to the training phase.

---
**Algorithm 1** Training procedure of GAP.

Let $T$ be total number of weight updates.
Let $N$ be the total number of GANs.
Let $K$ be the swapping frequency.
Let $\mathcal{M} = \{(G_1, D_1), (G_2, D_2), \cdots, (G_N, D_N)\}$.
**while** $t < T$ **do**
 Update $\mathcal{M}_{i_t} = (G_{i_t}, D_{i_t}) \, \forall i = 1 \cdots N$.

 **if** $t \, \% \, K == 0$ **then**
 Randomly select $\frac{N}{2}$ pairs with indices $(i, j)$ w/o replacement.
 Swap $D_i$ and $D_j$ ($G_i$ and $G_j$) $\forall i \neq j$.
 **end if**
**end while**
Select the best GAN based on GAM evaluation.

---

Our proposal trains multiple GANs simultaneously. However, unlike the popular method of data parallelism, we do not train them independently with shared parameters, rather we try to produce synergy effects among different GANs during the training phase. This can be achieved simply by randomly swapping different discriminators (generators) every $K$ updates. After training multiple GANS with our proposed method, we can select the best one based on the GAM. The pseudocode is shown in Algorithm 1.

We call our proposed method *generative adversarial parallelization* (GAP). Note that our method is not model-specific in a sense that GAP can be applied to any extension of GANs. For example, GAP can be applied to DCGAN or GRAN, or we can even apply GAP on several types of GANs simultaneously. Say, we have four GPUs available on which to parallelize models. We can allocate two GPUs for DCGANs and the remaining two GPUs for GRANs. Therefore, we view GAP as *an operator* rather than a model topology/architecture.

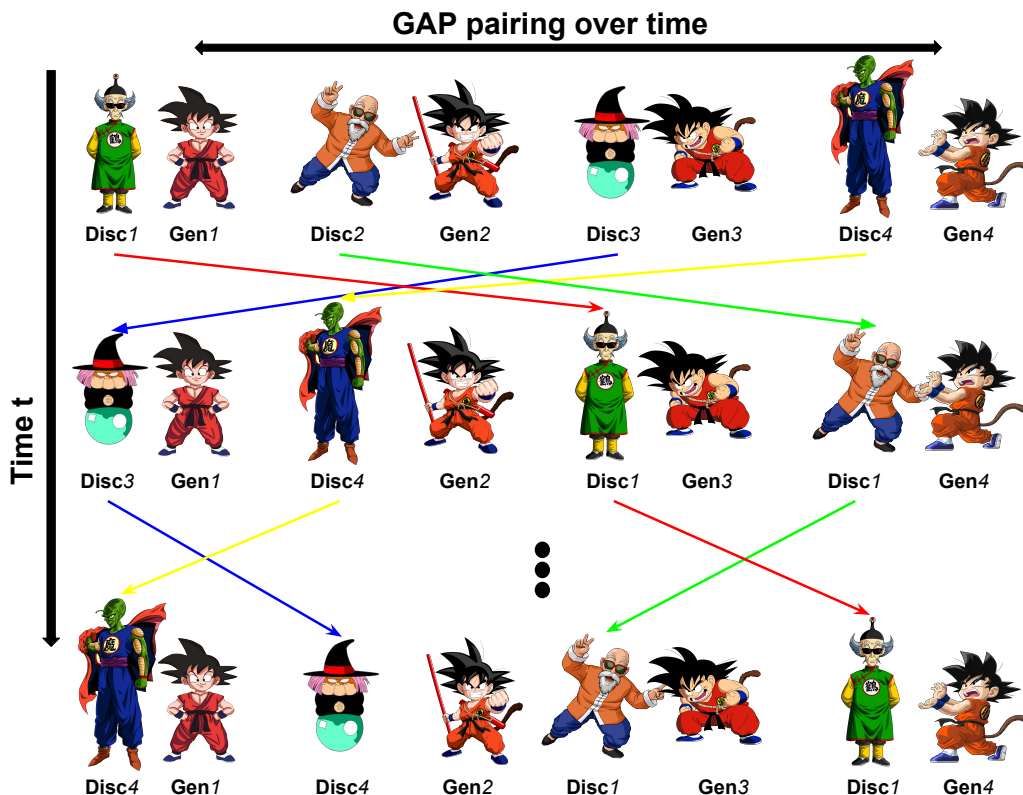

Figure 2: A cartoon illustration of Generative Adversarial Parallelization. Generators and discriminators are represented by different monks and sensei. The pairing between monks and sensei are randomly substituted overtime.

## 3.1 GAP AS REGULARIZATION

In a two player generative adversarial game, the concept of overfitting still exists. However, the realization of overfitting can be hard to notice. This is mainly due to not having a reconstructive error function. For models with a reconstruction-based objective, samples will simply become identical to the training data as the error approaches zero. On the other hand, with the GAN objective, even when the error approaches zero, it does not imply that the samples will look like the data. So, how can we characterize overfitting in a GAN?

We argue that overfitting in GANs manifests itself differently than in reconstructive models. Let us explain using two analogies to describe this phenomenon. Consider a generator as a judo fighter and discriminator as a sparring partner. When a judo fighter is only trained with the same sparring partner, his/her fighting strategy will naturally adapt to the style of his/her sparring partner. Thus, when the fighter is exposed to a new fighter with a different style, this fighter may suffer. Similarly, if a student learns from a single teacher, his/her learning experience will not only be limited but even overfitted to the teacher's style of exams (see Figure 2). Equivalently, a paired generator and discriminator are likely to be adapted to their own strategy. Here, GAP intrinsically prevents this problem as the generator (discriminator) periodically gets paired with different discriminator (generator). Thus, GAP can be viewed as a regularizer.

## 3.2 MODE COVERAGE

The kind of *overfitting problem* mentioned above further relates to the problem of assigning probability mass to different modes of the data generating distribution – what we call *mode coverage*.

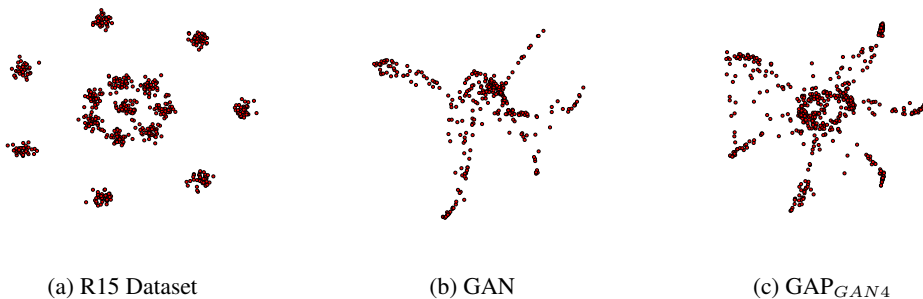

| (a) R15 Dataset | (b) GAN | (c) GAP$_{GAN4}$ |

Figure 3: a) The R15 dataset. Samples drawn from b) GAN and c) GAP$_{GAN4}$. GAP$_{GAP4}$ denotes four GANs trained in parallel with swapping at every epoch. The two models were trained using 100 out of 600 data points from the R15 dataset.

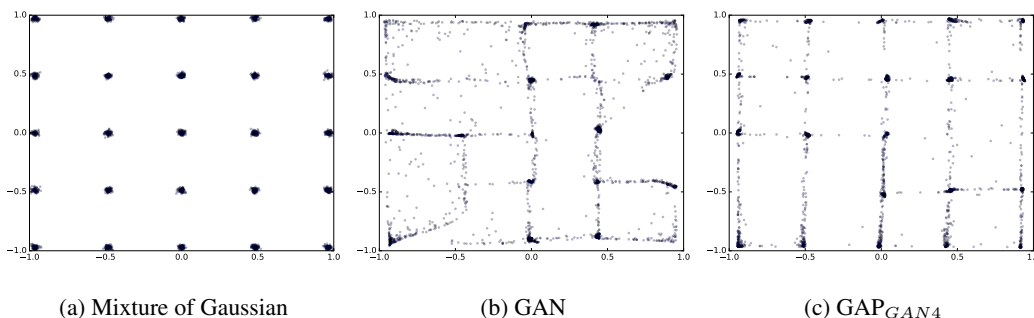

| (a) Mixture of Gaussian | (b) GAN | (c) GAP$_{GAN4}$ |

Figure 4: a) The Mixture of Gaussians dataset. Samples drawn from b) GAN and c) GAP$_{GAN4}$. GAP$_{GAP4}$ denotes four GANs trained in parallel with swapping at every epoch. The two models were trained using 2500 examples.

Let us re-consider the example introduced in Section 2.1. Say, the generator was able to figure out a single mode from which samples are drawn that confuse the discriminator. As long as the discriminator does not learn to fix this problem, the generator is not motivated to consider any other modes. This kind of scenario allows the generator to cheat by staying within a single, or small set of modes rather than exploring alternatives.

The story is not exactly the same when there are several different discriminators interacting with each generator. Since different discriminators may be good at distinguishing samples from different modes, each generator must put some effort into fooling all of the discriminators by generating samples from different modes. The situation where samples from a single mode fool all of the discriminators grows much less likely as the number and diversity of discriminators and generators increases (see Figure 3 and 4). Full details of this visualization are provided in Section 4.1.

## 4 EXPERIMENTS

We conduct an empirical investigation of GAP using two recently proposed GAN-variants as starting points: DCGAN (Radford et al., 2015) and GRAN (Im et al., 2016)[2]. In each case, we compare individual GAN-style models to GAP-style ensembles trained in parallel.

As it is difficult to quantitatively assess mode coverage, first we aim to visualize samples from GAP vs. other GAN variants on low-dimensional (toy) datasets as well as low-dimensional projections on real data. Then to evaluate each model quantitatively, we apply the GAM-II metric which is a re-formulation of GAM (Im et al., 2016) which can be used to compare different GAN architectures. Its motivation and use is described in Section 4.1. We consider, in total, five GAP variants which are summarized in Table 1.

---

[2] The Theano-based DCGAN and GRAN implementations were based on https://github.com/Newmu/dcgan and https://github.com/jiwoongim/GRAN, respectively.

Table 1: GAP variants and their short-hand labels considered in our experiments.

| Name | Model | Description |
|---|---|---|
| $GAP_{D2}$ | GAP(DCGAN×2) | Two DCGANs trained with GAP. |
| $GAP_{D4}$ | GAPDC(DCGAN×4) | Four DCGANs trained with GAP. |
| $GAP_{G2}$ | GAP(GRAN×2) | Two GRANs trained with GAP. |
| $GAP_{G4}$ | GAP(GRAN×4) | Four GRANs trained with GAP. |
| $GAP_{C4}$ | GAP(DCGAN×2, GRAN×2) | Two DCGANs and two GRANs trained with GAP. |

## 4.1 EXPERIMENTAL SETUP

All of our models are implemented in Theano (Bergstra et al., 2010) – a Python library that facilitates deep learning research. Because every update of each model is implemented as a separate process during training, swapping their parameters among different GANs necessitates interprocess communication[3]. Similar to the Theano-MPI framework, we chose to do inter-GPU memory transfer instead of passing through host memory in order to reduce communication overhead. Random swapping of the two discriminators' parameters is achieved with an in-place `MPI_SendRecv` operation as DCGAN and GRAN share the same architecture and therefore the same parameterization.

Throughout the experiments, all datasets were normalized between $[0, 1]$. We used the same hyperparameters reported in (Radford et al., 2015) and (Im et al., 2016) for DCGAN and GRAN, respectively. The only additional hyper-parameter introduced by GAP is the frequency of swapping discriminators during training. We also made deliberate fine-grained distinctions among each GAN trained under GAP. These were: i) the generator's prior distribution was selected as either uniform or Gaussian; ii) the order of mini-batches was permuted during learning; and iii) noise was injected at the input during learning and the amount of noise was decayed over time. The point of introducing these distinctions was to avoid multiple GANs converging to the same or very similar solutions. Lastly, we used gradient clipping (Pascanu et al., 2013) on both discriminators and generators.

To measure the performance of GANs, our first attempt was to apply GAM to evaluate our model. Unfortunately, we realized that GAM is not applicable when comparing GAP vs. non-GAP models. This is because GAM requires the discriminator the GANs under comparison to have similar error rates on a held-out test set. However, as shown in Figure 6, GAP boosts the generalization of the discriminators, which causes it to have different test error rates compared to the error rate from non-GAP models. Hence, we propose a new metric that omits the GAM's constraints which we call GAM-II. It simply measures the average (or worst case) error rate among a collection of discriminators. A detailed description of GAM-II is provided in Appendix A.1.

## 4.2 RESULTS

We report our experimental results by answering a few core questions.

*Q: Do GAP-trained models cover more modes of the data generating distribution?*

Determining whether applying GAP achieves broader mode coverage is difficult to validate in high-dimensional spaces. Therefore, we initially verified GAP and non-GAP models on two low-dimensional synthetic datasets. The R15 dataset[4] contains 500 two-dimensional data points with 15 clusters as shown in Figure 3a. The Mixture of Gaussians dataset[5] contains 2,500 two-dimensional data points with 25 clusters as shown in Figure 4a.

Both discriminator and generator had four fully-connected batch-normalized layers with ReLU activation units. We first optimized the hyper-parameters of a single GAN based on visually inspecting the samples that it generated (i.e. Figure 3 shows samples from the best performing single GAN that we trained). We then trained four parallelized GANs using the same hyper-parameters of the best single GAN.

The samples generated from both models are shown in Figure 3 and 4. We observe that GAP(GAN×4) produces samples that look more similar to the original dataset compared to a single

---

[3]We used openMPI for implementing GAP – see https://www.open-mpi.org/

[4]The R15 dataset can be found at https://cs.joensuu.fi/sipu/datasets/

[5]The Mixture of Gaussians dataset can be found at https://github.com/IshmaelBelghazi/ALI/

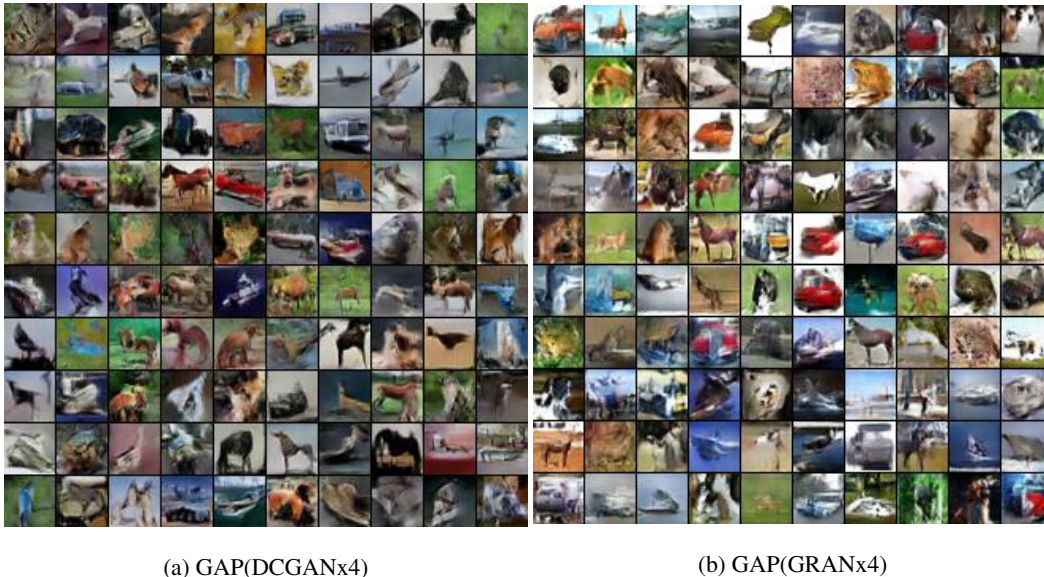

(a) GAP(DCGANx4)  (b) GAP(GRANx4)

Figure 5: CIFAR-10 samples. Best viewed in colour. More samples are provided in the Appendix.

GAN. The overlap of samples generated by four GANs are consistent with Figure 3c. Note that as we decrease the number of training points, the overlap of GAN samples deviates from the original dataset while GAP seems not to suffer from this phenomenon. For example, when we used all 600 examples of R15, both GAN and GAP samples matched the distribution of data in Figure 3a. However, as we use less training examples, GAN failed to accurately model the data distribution by dropping modes. The samples plotted in Figure 3c are based on training each model with a random subset of 100 examples drawn from the original 600. Based on the synthetic experiments we confirm that GAP can improve mode coverage when a limited number of training samples are available.

In order to gain a qualitative sense of models trained using a high dimensional dataset, we considered two experiments: i) we examined the class label predictions made on samples from each model to check how uniformly they were distributed. The histogram of the predicted classes is provided in Figure 14. ii) we created a t-SNE visualization of generated samples overlaid on top of the true data (see Appendix A.2). We find that the intersection of data points and samples generated by GAP is slightly better than samples generated by individual GANs. In addition to the synthetic data results, these visualizations suggest some favourable properties of GAP, but we hesitate to draw any strong conclusions from them.

*Q: Does GAP enhance generalization?*

To answer this question, we considered the MNIST, CIFAR-10, and LSUN church datasets which are often used to evaluate GAN variants. MNIST and CIFAR-10 consist of 50,000 training and 10,000 test images of size $27 \times 28$ and $32 \times 32 \times 3$ pixels, respectively. Each contains 10 different classes of objects. The LSUN church dataset contains various outdoor church images. These high resolution images were downsampled to $64 \times 64$ pixels. The training set consists of 126,227 examples.

One implicit but imperfect way to measure the generalization of a GAN is to observe generalization of the discriminator alone. This is because the generator is influenced by the discriminator and vice versa. If the discriminator is overfitting the training data, then the generator must be biased towards the training data as well. Here, we plot the learning curve of the discriminator during training for both GAP(DCGAN) and GAP(GRAN).

Figure 6 shows the learning curve for a single model versus groups of two and four models parallelized under GAP. We observe that more parallelization leads to less of a spread between the train and validation curves indicating the ability of GAP to improve generalization. Note that in order to plot a single representative learning curve while training multiple models under GAP, we averaged the learning curves of the individual models. To demonstrate that our observations are not merely attributable to smoothing by averaging, we show individual learning curves of the parallelized GANs (see Figure 13 in Appendix A.3). From now on, we will work with $GAP_{D4}$ and $GAP_{G4}$.

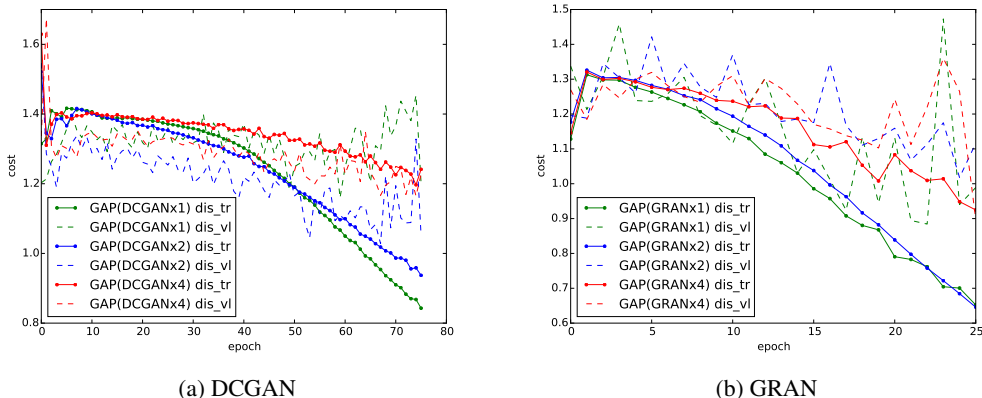

(a) DCGAN  (b) GRAN

Figure 6: Discriminator learning curves on CIFAR-10 as a proxy for generalization performance. As parallelization scales up, the spread between training and validation cost shrinks. Note that the curves corresponding to "GAP(DCGANx2)", "GAP(DCGANx4)", "GAP(GRANx2)" and "GAP(GRANx4)" are averages of the corresponding GAP models. See Figure 13 in Appendix A.3 for the individual curves before averaging.

*Q: How does the rate at which discriminators are swapped affect training?*

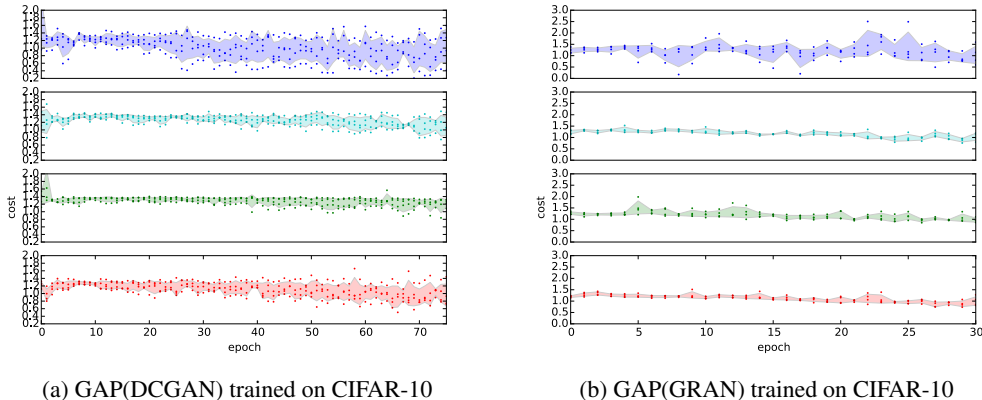

(a) GAP(DCGAN) trained on CIFAR-10  (b) GAP(GRAN) trained on CIFAR-10

Figure 7: The standard deviations of the validation costs at various swapping frequencies. From top to bottom: 0.1, 0.3, 0.5, 0.7, and 1.0 per epoch.

As noted earlier, the swapping frequency is the only additional hyper-parameter introduced by GAP. We conduct a simple sensitivity analysis by plotting the validation cost of each GAN during training along with its standard deviation in Figure 7. We observe that GAP(DCGAN) varies the least at a swapping frequency of 0.5 – swapping twice per epoch. Meanwhile, GAP(GRANs) are not too sensitive to swapping frequencies above 0.1. Figure 12 in Appendix A.3 plots learning curves at different swapping frequencies. Across all rates, we still see that the spread between the training and validation costs decreases with the number of GANs trained in parallel.

*Q: Does GAP(·) improve the quality of generative models?*

We used GAM-II to evaluate GAP (see Appendix A.1). We first looked at the performance over four models: DCGAN, GRAN, $GAP_{D4}$, and $GAP_{G4}$. We also considered combining multiple GAN-variants in a GAP model (hybrid GAP). We denote this model as $GAP_{C4}$. $GAP_{C4}$ consists of two DCGANs and two GRANs trained with GAP. Overall, we have ten generators and ten discriminators for DCGAN and GRAN: four discriminators from the individually-trained models, and four discriminators from GAP, and two discriminators from the GAP combination, $GAP_{C4}$. We used the collection of all ten discriminators to evaluate the generators. Table 3 presents the results. Note that we report the minimum and maximum of average and worst error rates among four GANs. Looking

at the average errors, $GAP_{D4}$ strongly outperforms DCGAN on all datasets. $GAP_{G4}$ outperforms GRAN on CIFAR-10 and MNIST and strongly outperforms it on LSUN. For the case of the maximum worst-case error, GAP outperforms both DCGAN and GRAN across all datasets. However, we did not find an improvement on $GAP_{C4}$ based on the GAM-II metric.

Additionally, we estimated the log-likelihood assigned by each model based on a recently proposed evaluation scheme that uses Annealed Importance Sampling (Wu et al., 2016). With the code provided by (Wu et al., 2016), we were able to evaluate DCGANs trained by $GAP_{D4}$ and $GAP_{comb}$[6]. The results are shown in Table 4. Again, these results show that $GAP_{D4}$ improves on DCGAN's performance, but there is no advantage using combined $GAP_{C4}$.

Samples from each CIFAR-10 and LSUN model for visual inspection are reproduced in Figures 16, 17, 18, and 19.

Table 2: DCGANs versus GAP(DCGAN) evaluation using GAM-II.

| DATASET | MODELS | DCGAN | | $GAP_{D4}$ | | $GAP_{C4}$ | |
|---|---|---|---|---|---|---|---|
| | MEASURE | MIN | MAX | MIN | MAX | MIN | MAX |
| MNIST | AVG. | 0.352 | 0.395 | 0.430 | 0.476 | 0.398 | 0.423 |
| | WORST | 0.312 | 0.351 | 0.355 | 0.405 | 0.326 | 0.343 |
| CIFAR-10 | AVG. | 0.333 | 0.368 | 0.526 | 0.565 | 0.888 | 0.902 |
| | WORST | 0.173 | 0.225 | 0.174 | 0.325 | 0.551 | 0.615 |
| LSUN | AVG. | 0.592 | 0.628 | 0.619 | 0.652 | 0.108 | 0.180 |
| | WORST | 0.039 | 0.078 | 0.285 | 0.360 | 0.0 | 0.0 |

Table 3: GRAN versus GAP(GRAN) evaluation using GAM-II.

| DATASET | MODELS | GRAN | | $GAP_{G4}$ | | $GAP_{C4}$ | |
|---|---|---|---|---|---|---|---|
| | MEASURE | MIN | MAX | MIN | MAX | MIN | MAX |
| MNIST | AVG. | 0.433 | 0.465 | 0.510 | 0.533 | 0.459 | 0.474 |
| | WORST | 0.004 | 0.020 | 0.008 | 0.020 | 0.010 | 0.012 |
| CIFAR-10 | AVG. | 0.289 | 0.355 | 0.332 | 0.416 | 0.306 | 0.319 |
| | WORST | 0.006 | 0.019 | 0.048 | 0.171 | 0.001 | 0.023 |
| LSUN | AVG. | 0.477 | 0.590 | 0.568 | 0.649 | 0.574 | 0.636 |
| | WORST | 0.013 | 0.043 | 0.022 | 0.055 | 0.015 | 0.021 |

Table 4: The likelihood of DCGANs and GAP(DCGAN) using the AIS estimator proposed by (Wu et al., 2016) on the MNIST dataset.

| MODELS | DCGAN | $GAP_{D4}$ |
|---|---|---|
| AIS | $682.5 \pm 12.51$ | $691.6 \pm 0.01$ |

## 5 DISCUSSION

We have proposed Generative Adversarial Parallelization, a framework in which several adversarially-trained models are trained together, exchanging discriminators. We argue that this reduces the tight coupling between generator and discriminator and show empirically that this has a beneficial effect on mode coverage, convergence, and quality of the model under the GAM-II metric. Several directions of future investigation are possible. This includes applying GAP to the evolving variety of adversarial models, like improvedGAN (Salimans et al., 2016). We still view stability as an issue and partially address it by tricks such as clipping the gradient of the discriminator. In this work, we only explored synchronous training of GANs under GAP, however, asynchronous training may provide more stability. Recent work has explored the connection between GANs and actor-critic methods in reinforcement learning (Pfau & Vinyals, 2016). Under this view, we believe that GAP may have interesting implications for multi-agent RL. Although we have assessed mode coverage qualitatively either directly or indirectly via projections, quantitatively assessing mode coverage for generative models is still an open research problem.

---

[6]Unfortunately, we did not get the code provided by (Wu et al., 2016) to work on GRAN.

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

## A    Supplementary Material for Generative Adversarial Parallelization

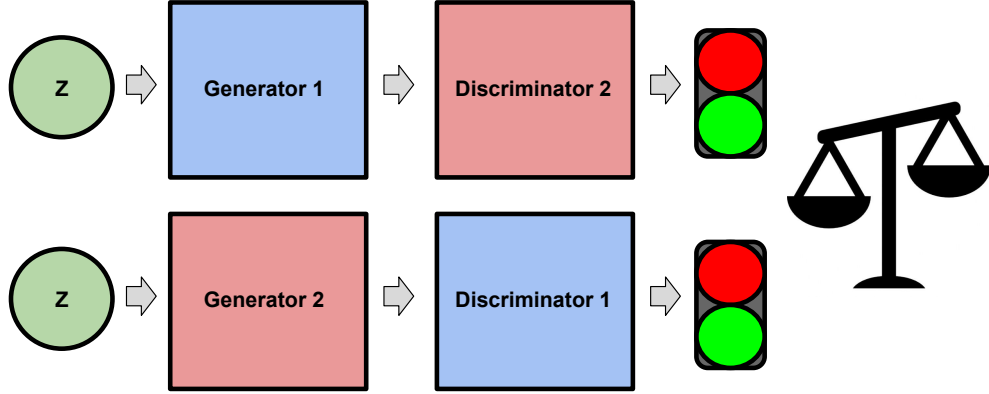

Figure 8: Illustration of the Generative Adversarial Metric.

### A.1    Generative Adversarial Metric II

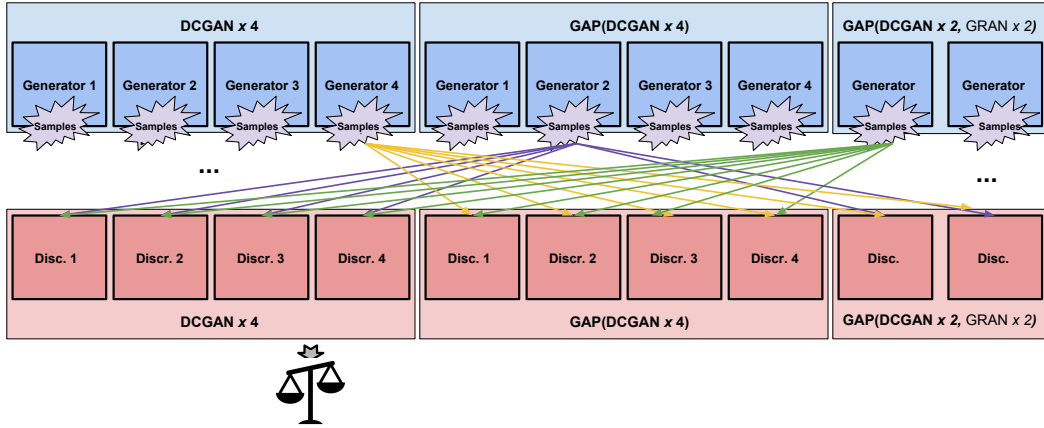

Figure 9: Illustration of the Generative Adversarial Metric II.

Although GAM is a valid metric as it measures *the likelihood ratio* of two generative models, it is hard to apply in practice. This is due to having a test ratio constraint, which imposes the condition that the ratio between test error rates be approximately unity. However, because GAP improves the generalization of GANs as shown in Figure 6, the test ratio often does not equal one (see Section 4). We introduce a new generative adversarial metric, and call it GAM-II.

GAM-II evaluates a model based on either the average error rate or worst error rate of a collection of discriminators given a set of samples from each model to be evaluated:

$$\underset{\{G_j|S_j\sim p_{\mathcal{G}_j}\}}{\operatorname{argmax}} \ \hat{\epsilon}(S_j) = \underset{\{G_j|S_j\sim p_{\mathcal{G}_j}\}}{\operatorname{argmax}} \ \frac{1}{N_j}\sum_{i=1}^{N_j}\epsilon(S_j|D_i), \tag{3}$$

$$\underset{\{G_j|S_j\sim p_{\mathcal{G}_j}\}}{\operatorname{argmax}} \ \bar{\epsilon}(S_j) = \underset{\{G_j|S_j\sim p_{\mathcal{G}_j}\}}{\operatorname{argmax}} \ \min_{i=1\cdots N_j}\epsilon(S_j|D_i) \tag{4}$$

where $\epsilon$ outputs the classification error rate, and $N_j$ is all discriminators except for the ones that the generator $j$ saw during training. For example, the comparison of DCGAN and GAP applied to four DCGANs is shown in Figure 9.

**Definition.** *We say that GAP helps if at least one of the models trained with GAP performs better than a single model. Moreover, GAP strongly-helps if all models trained with GAP perform better than a single model.*

In our experiments, we assess GAP based on the definition above.

## A.2 EXPERIMENTS WITH T-SNE

In order to get a qualitative sense of models trained using a high dimensional dataset, we consider a t-SNE map of generated samples overlaid on top of the true data. Normally, a t-SNE map is used to visualize clusters of embedded high-dimensional data. Here, we are more interested in the overlap between true data and generated samples by visualizing clusters which we interpret as modes of the data generating distribution.

Figure 10 and 11 present the t-SNE map of data and samples from single- and multiple- trained GANs under GAP. We find that the intersection of data points and samples generated by GAP is slightly better than samples generated by individual GANs. This provides an incomplete view but is nevertheless a helpful visualization.

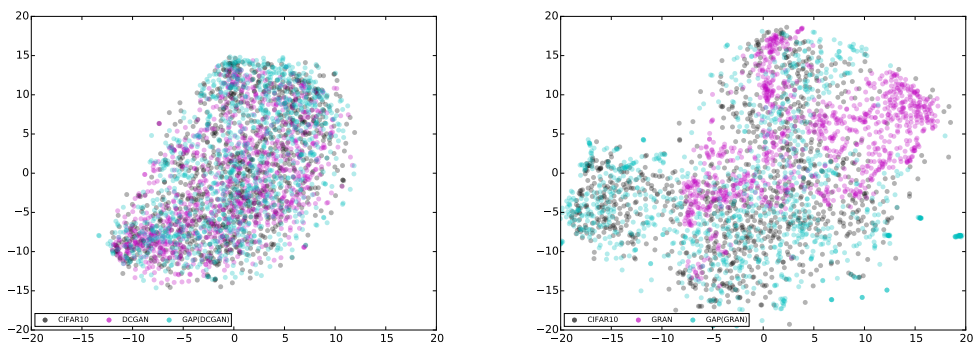

(a) Using data, DCGAN, & GAP(DCGAN) samples   (b) Using data, GRAN, & GAP(GRAN) samples

Figure 10: t-SNE mapping of data and sample points on CIFAR-10. The points are colour coded as: Data (Black), Single Model (Magenta), and GAP (Cyan). Note that, particularly for the figure on the right, there seems to be more overlap between the data and the GAP-generated samples compared to the GAN-generated samples.

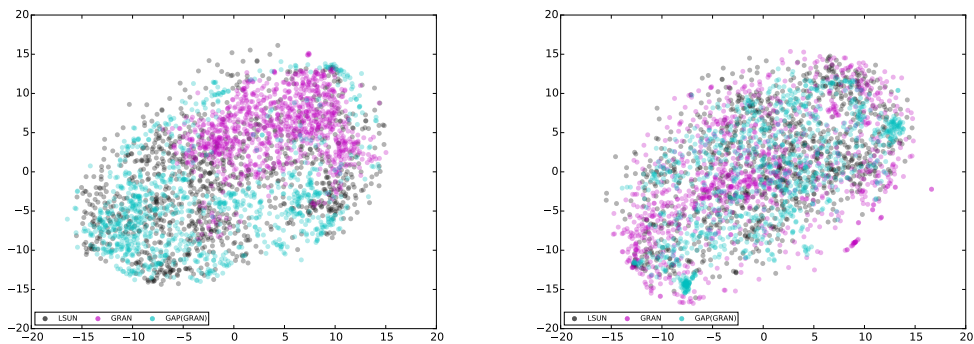

(a) Using data, DCGAN, & GAP(DCGAN) samples   (b) Using data, GRAN, & GAP(GRAN) samples

Figure 11: t-SNE mapping of data and sample points on the LSUN dataset. The points are colour coded as: Data (Black), Single Model (Magenta), and GAP (Cyan).

A.3 SUPPORTING FIGURES

The supporting figures as in Figure 6 is presented for LSUN dataset in Figure 12. There are total of four plots with different swapping frequencies.

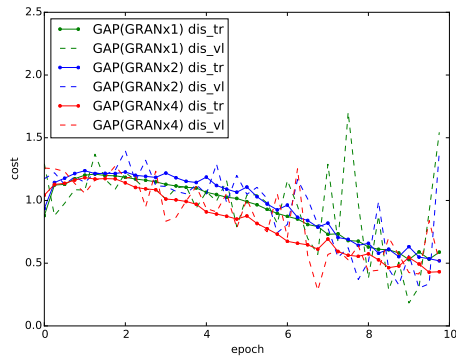

(a) swapping frequency every 0.1 epoch.

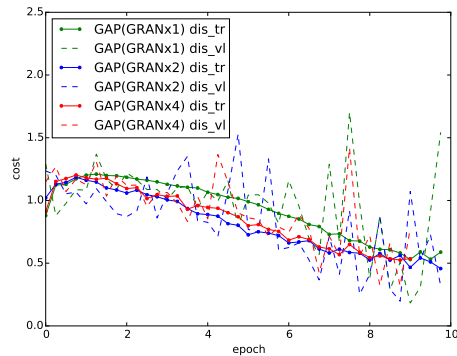

(b) swapping frequency every 0.3 epoch.

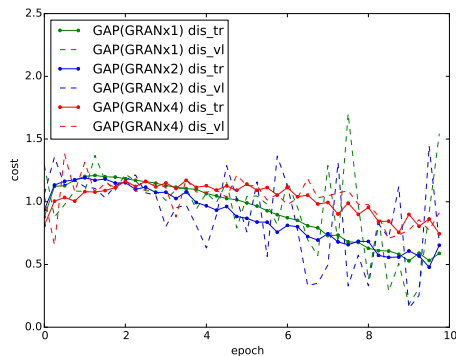

(c) swapping frequency every half epoch.

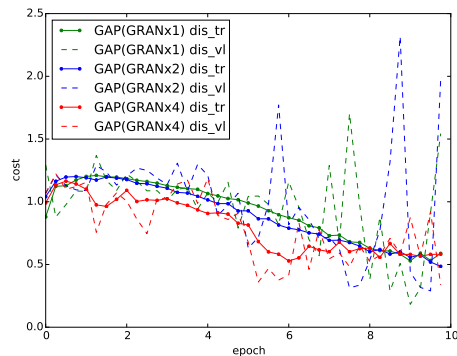

(d) swapping frequency every epoch.

Figure 12: Averaged GAP(GRAN) learning curves trained on the LSUN Church dataset. As parallelization scales up, the gap between training and validation cost narrows.

Figure 13 presents an instance of an individual learning curve in the case when multiple GANs are trained under GAP. The difference from from Figure 6 and Figure 12 is that GAP curves are represented by the learning curve of a single GAN within GAP rather than an average. Fortunately, the behaviour remains the same, where the spread between training and validation cost decreases as parallelization scales up (i.e. more models in a GAP).

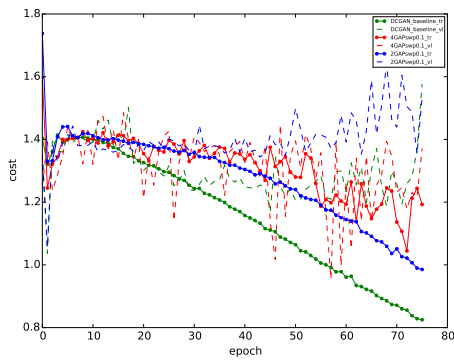

(a) Swapping frequency every 0.1 epoch.

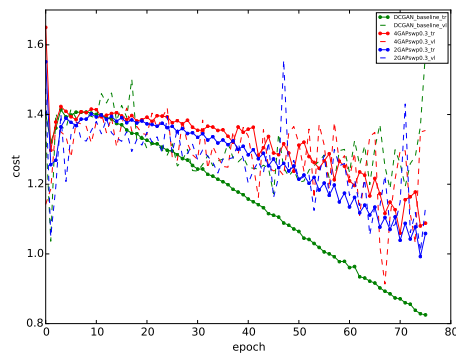

(b) Swapping frequency every 0.3 epoch.

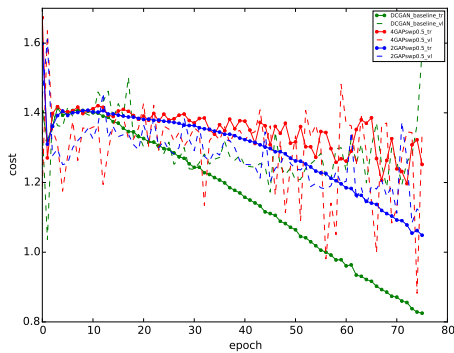

(c) Swapping frequency every half epoch.

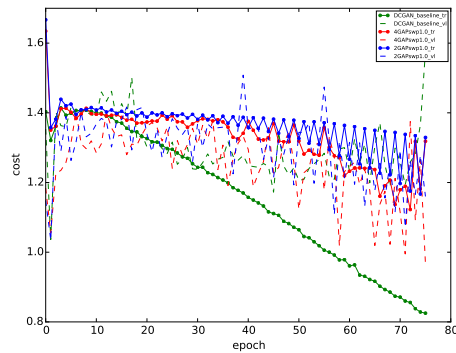

(d) Swapping frequency every epoch.

Figure 13: Each GAP model represented by the learning curve of a single DCGAN within the GAP(DCGAN) trained on CIFAR-10. This demonstrates that the observed behaviour of reducing the spread between training and validation cost is not simply an effect of averaging.

We observed the distribution of class predictions on samples from each model in order to check how closely they match the training set distribution (which is uniform for MNIST). We trained a simple logistic regression on MNIST that resulted in a $\simeq 99\%$ test accuracy rate. The histogram of the predicted classes is provided in Figure 14. We looked at the exponentiated expected KL divergence between the predicted distribution and the (uniform) prior distribution, also known as the "Inception Score" (Salimans et al., 2016). The results are shown in Table 5.

Table 5: Inception Score of GAP

| MODELS | DCGAN | GAP$_{D4}$ | GRAN | GAP$_{G4}$ |
|---|---|---|---|---|
| MNIST | $6.36 \pm 0.09$ | $6.592 \pm 0.08$ | $6.77 \pm 0.08$ | $6.85 \pm 0.24$ |

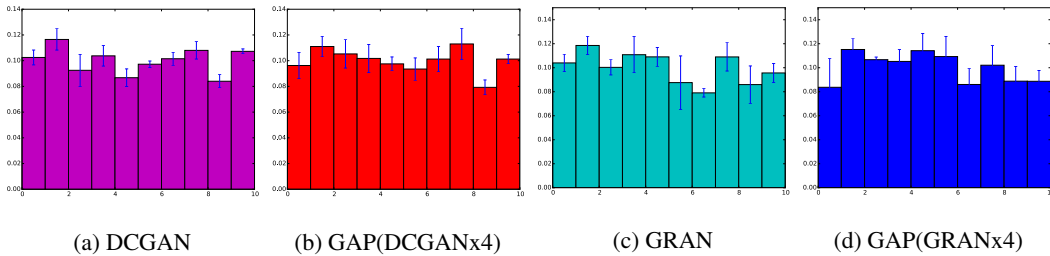

(a) DCGAN          (b) GAP(DCGANx4)          (c) GRAN          (d) GAP(GRANx4)

Figure 14: The distribution of the predicted class labels of samples from various models made by a separately trained logistic regression.

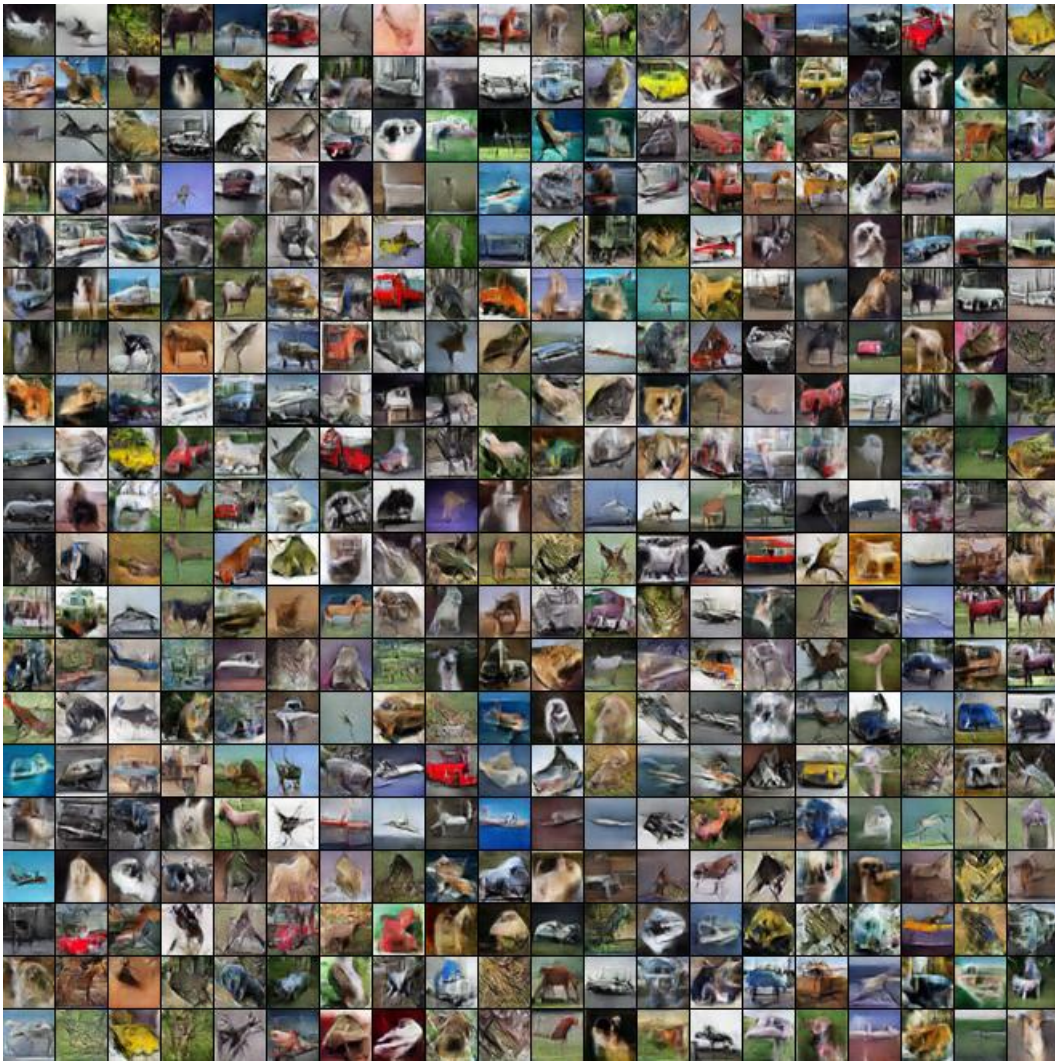

Figure 15: CIFAR-10 samples generated by GAP[DCGANx4]. Best viewed in colour.

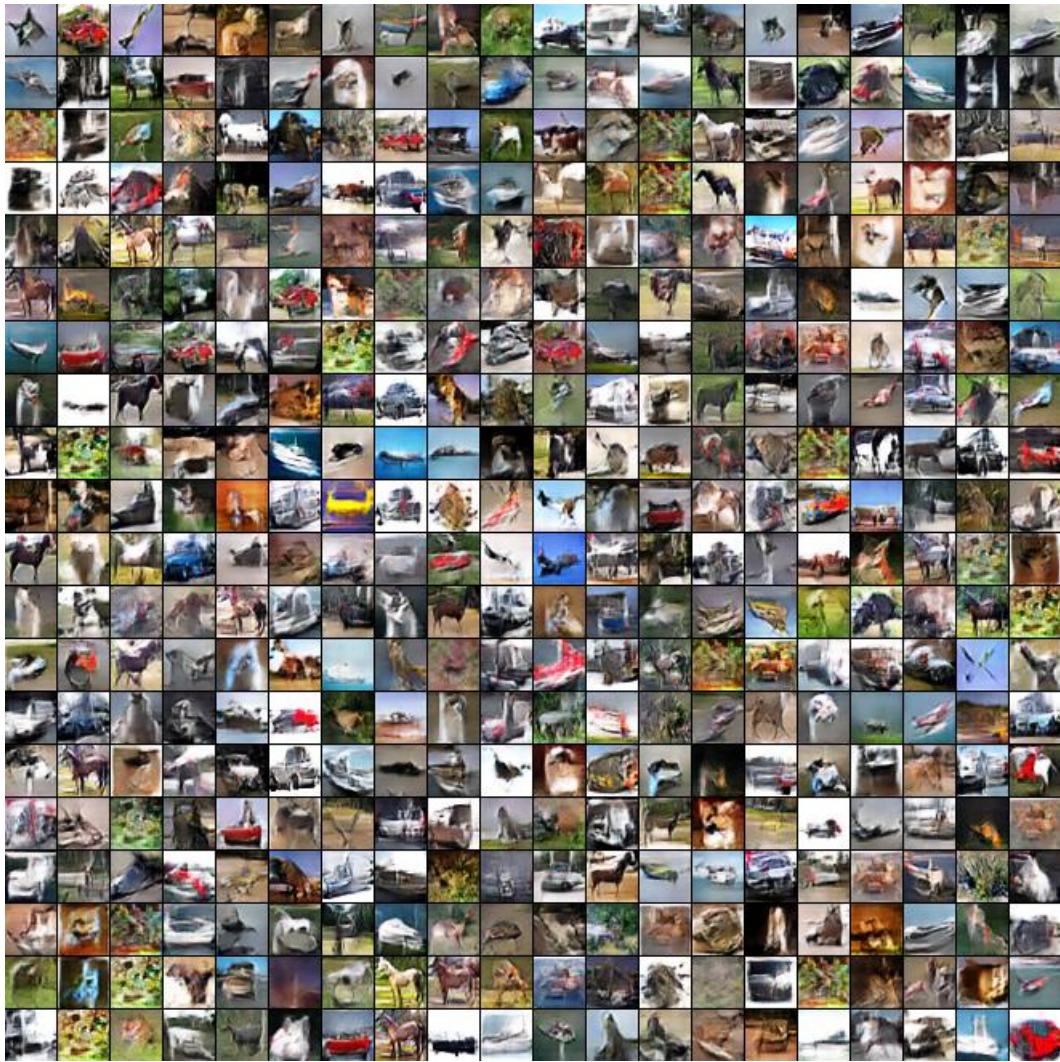

Figure 16: CIFAR-10 samples generated by GAP[GRANx4]. Best viewed in colour.

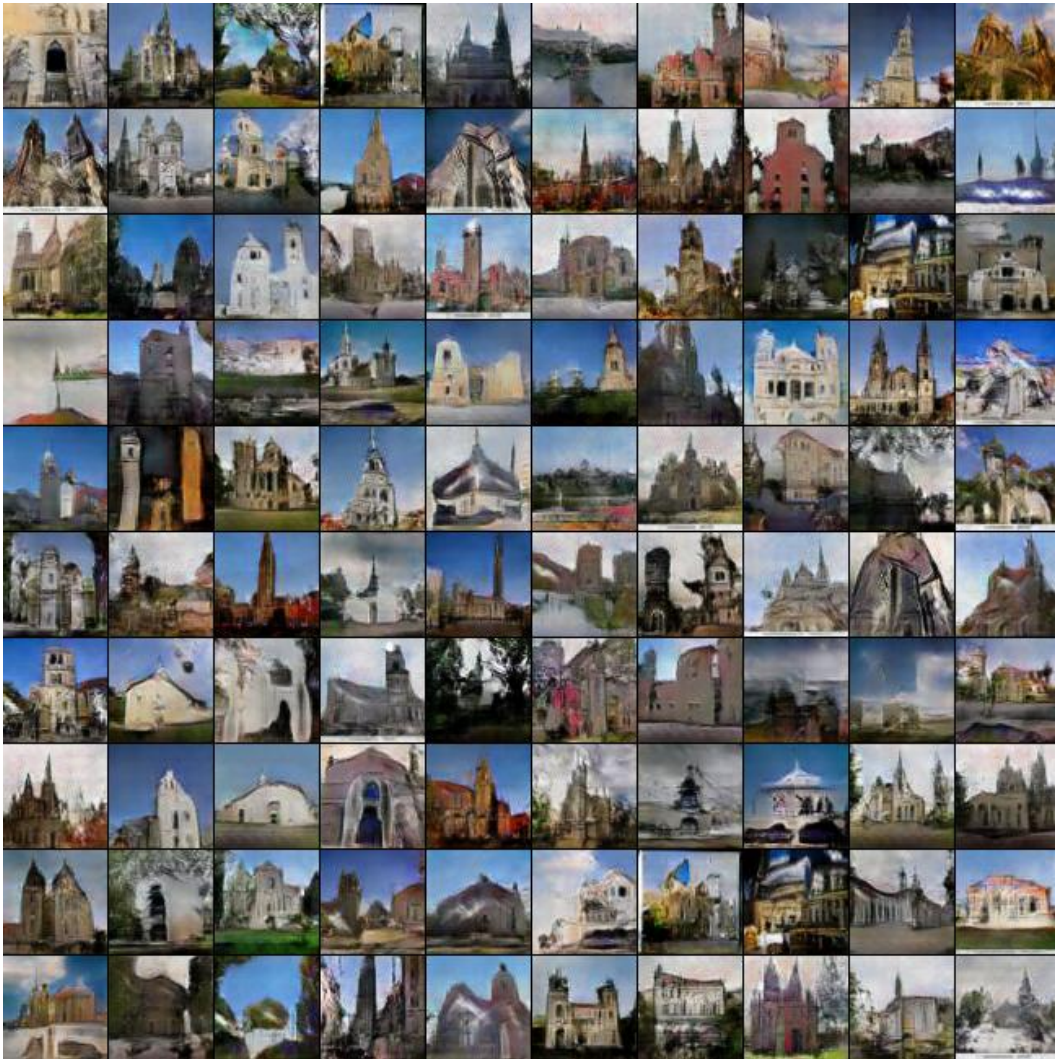

Figure 17: LSUN Church samples generated by GAP[DCGANx4] at 0.3 swapping frequency. Best viewed in colour.

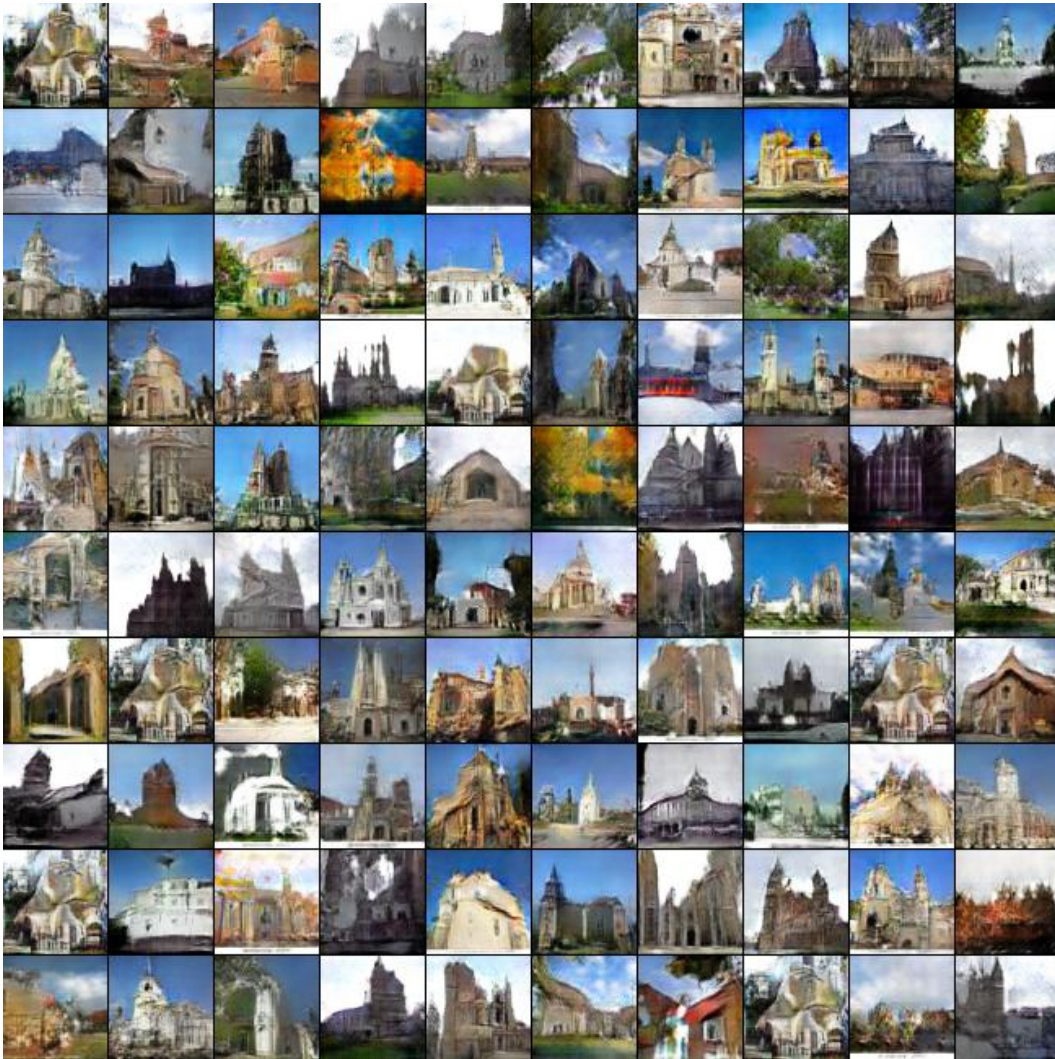

Figure 18: LSUN Church samples generated by GAP[GRANx4] at 0.5 swapping frequency. Best viewed in colour.

### A.3.1 FINE-TUNING GANs USING GAP

We also tried fine-tuning individually-trained GANs using GAP, which we denote as $GAP_{F4}$. $GAP_{F4}$ consists of two trained DCGANs and two trained GRANs. They are then fine-tuned using GAP for five epochs. Samples from the fine-tuned models are shown in Figure 20.

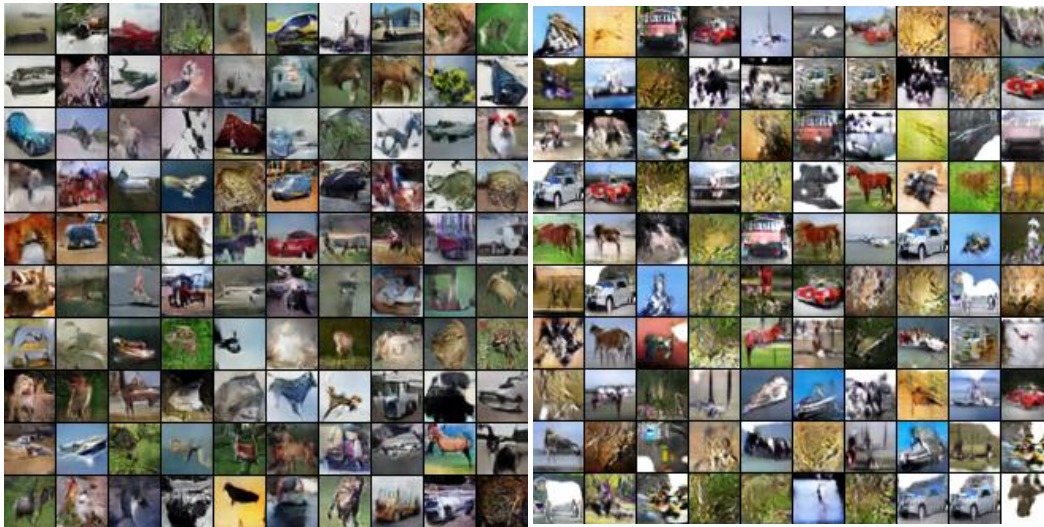

(a) Samples from one of the DCGANs trained using $GAP_{C4}$

(b) Samples from one of the GRANs trained using $GAP_{C4}$

Figure 19: CIFAR-10 samples trained by GAP(DCGANx2, GRANx2).

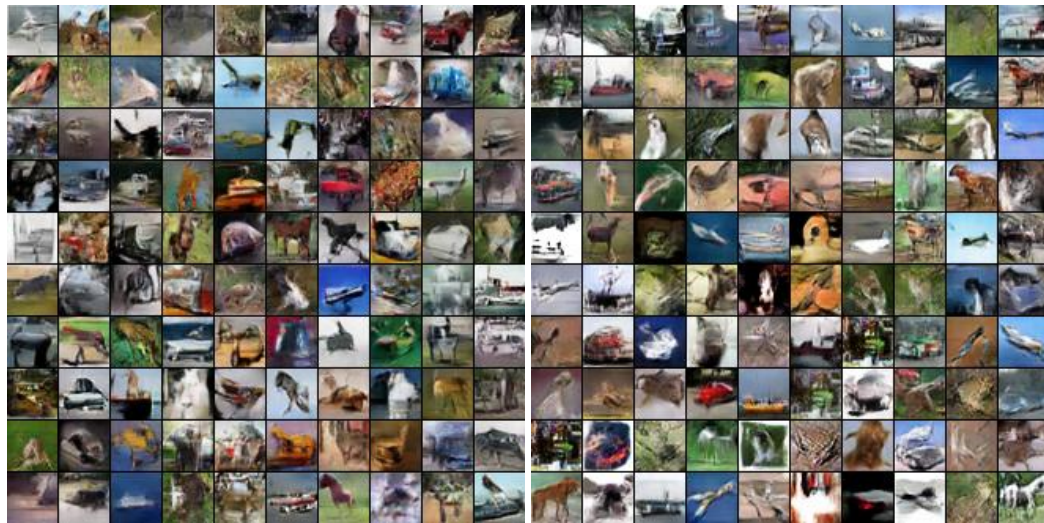

(a) Samples from one of the fine-tuned DCGAN using $GAP_{C4}$

(b) Samples from one of the fine-tuned GRAN using $GAP_{C4}$

Figure 20: CIFAR-10 samples trained by GAP(DCGANx2, GRANx2).

