# Peer review of "Generative Adversarial Parallelization"

_ICLR 2017 — rejected_

[Official Review · AnonReviewer1 · rating 4 · confidence 4 · 13 Dec 2016]
**No Title**

This paper proposes Generative Adversarial Parallelization (GAP), one schedule to train N Generative Adversarial Networks (GANs) in parallel. GAP proceeds by shuffling the assignments between the N generators and the N discriminators at play every few epochs. Therefore, GAP forces each generator to compete with multiple discriminators at random. The authors claim that such randomization reduces undesired "mode collapsing behaviour", typical of GANs.

I have three concerns with this submission.

1) After training the N GANs for a sufficient amount of time, the authors propose to choose the best generator using the GAM metric. I oppose to this because of two reasons. First, a single GAN will most likely be unable to express the full richness of the true data begin modeled. Said differently, a single generator with limited power will either describe a mode well, or describe many modes poorly. Second, GAM relies on the scores given by the discriminators, which can be ill-posed (focus on artifacts). Since there is There is nothing wrong with mode collapsing when this happens under control. Thus, I believe that a better strategy would be to not choose and combine all generators into a mixture. Of course, this would require a way to decide on mixture weights. This can be done, for instance, using rejection sampling based on discriminator scores.

2) The authors should provide a theoretical (or at least conceptual) comparison to dropout. In essence, this paper has a very similar flavour: every generator is competing against all N discriminators, but at each epoch we drop N-1 for every generator. Related to the previous point, after training dropout keeps all the neurons, effectively approximating a large ensemble of neural networks.

3) The qualitative results are not convincing. Most of the figures show only results about GAP. How do the baseline samples look like? The GAN and LAPGAN papers show very similar samples. On the other hand, I do not find Figures 3 and 4 convincing: for instance, the generator in Figure 3 was most likely under-parametrized.

As a minor comment, I would remove Figure 2. This is because of three reasons: it may be protected by copyright, it occupies a lot of space, and it does not add much value to the explanation. Also, the indices (i_t) are undefined in Algorithm 1.

Overall, this paper shows good ideas, but it needs further work in terms of conceptual development and experimental evaluation.

[Official Review · AnonReviewer2 · rating 4 · confidence 4 · 16 Dec 2016]

This paper proposes to address the mode collapsing problem of GANs by training a large set of generators and discriminators, pairing them each up with different ones at different times throughout training. The idea here is that no one generator-discriminator pair can be too locked together since they are all being swapped. This idea is nice and is addressing an important issue with GAN training. However, I think the paper is lacking in experimental results. In particular:

- The authors need to do more work to motivate the GAM metric. It is not intuitively obvious to me that the GAM metric is a good way of evaluating the generator networks since it relies on the prediction of the discriminator networks which can fixate on artifacts. Perhaps the authors could explore if the GAM metric correlates with inception scores or human evaluations. Currently the only quantitative evaluation uses this criterion and it really isn't clear it's a relevant quantity to be measuring. 
- Related to the above comment, the authors need to compare more to other methods. Why not evaluate inception scores and compare with previous methods. Similarly, generation quality is not compared with previous methods. It's not obvious that the sample quality is any better with this method. 

And now just repeating questions from pre-review section:
- If, instead of swapping, you were to simply train K GANs on K splits of the data, or K GANs with differing initial conditions (but without swapping) do you see any improvement in results? Similarly, how about if you train larger capacity models with dropout in G and D? Since dropout essentially averages many models it would be interesting to see if the effects are the same.
- In figure 6 it appears that the validation costs remain the same as parallelization increase, but the training cost goes up and that is why the gap is shrinking. Does this really imply better generalization?

In summary, interesting paper that addresses an important issue with GAN training, but compelling results are missing.

[Official Review · AnonReviewer3 · rating 4 · confidence 3 · 18 Dec 2016]

This paper proposes an extension of the GAN framework known as GAP whereby multiple generators and discriminators are trained in parallel. The generator/discriminator pairing is shuffled according to a periodic schedule.

Pros:
+ The proposed approach is simple and easy to replicate.

Cons:
- The paper is confusing to read.
- The results are suggestive but do not conclusively show a performance win for GAP.

The main argument of the paper is that GAP leads to improved convergence and improved coverage of modes. The coverage visualizations are suggestive but there still is not enough evidence to conclude that GAP is in fact improving coverage. And for convergence it is difficult to assess the effect of GAP on the basis of learning curves. The proposed GAM-II metric is circular in that model performance depends on the collection of baselines the model is being compared with. Estimating likelihood via AIS seems to be a promising way to evaluate, as does using the Inception score.

Perhaps a more systematic way to determine GAP's effect would be to set up a grid search of hyperparameters and train an equal number of GANs and GAP-GANs for each setting. Then a histogram over final Inception scores or likelihood estimates of the trained models would help to show whether GAP tended to produce better models. Overall the approach seems promising but there are too many open questions regarding the paper in its current form.

* Section 2: "Remark that when..." => seems like a to-do.
* Section A.1: The proposed metric is not described in adequate detail.

[Final Decision · Program Chairs · 06 Feb 2017]
**ICLR committee final decision**

This paper was reviewed by three experts. While they find interesting ideas in the manuscript, all three point to deficiencies (problems with the use of GAM metric, lack of convincing results) and unanimously recommend rejection. I do not see a reason to overturn their recommendation.